# Genome and Pathogenicity Analysis of an NADC30-like PRRSV Strain in China’s Xinjiang Province

**DOI:** 10.3390/v17030379

**Published:** 2025-03-06

**Authors:** Honghuan Li, Wei Zhang, Yanjie Qiao, Wenxing Wang, Wenxiang Zhang, Yueli Wang, Jihai Yi, Huan Zhang, Zhongchen Ma, Chuangfu Chen

**Affiliations:** 1College of Animal Science and Technology, Shihezi University, Shihezi 832003, China; lhh121004@126.com (H.L.); 18899596250@163.com (Y.Q.); 13103723620@163.com (W.W.); z302729814@163.com (W.Z.); 15899292491@163.com (J.Y.); Zhanghuan901212@126.com (H.Z.); 2College of Veterinary Medicine, Xinjiang Agricultural University, Urumqi 830052, China; zhangwei@tecon-bio.com; 3Tecon Bio-Pharmaceuticals Co., Ltd., Urumqi 830011, China; 4College of Medicine, Shihezi University, Shihezi 832008, China; wangyueli0710@163.com; 5Collaborative Innovation Center for Sheep Health Breeding and Zoonosis Prevention and Control, Shihezi 832003, China

**Keywords:** porcine reproductive and respiratory syndrome virus, NADC30-like strain, phylogenetic analysis, recombination analysis, pathogenicity

## Abstract

The porcine reproductive and respiratory syndrome virus (PRRSV) possesses an inherent ability to adapt to environmental transformations and undergo evolutionary changes, which has imposed significant economic pressure on the global pig industry. Given the potential for recombination among PRRSV genomes and variations in pathogenicity, newly emerging PRRSV isolates are of considerable clinical importance. In this study, we successfully isolated a novel strain named XJ-Z5 from PRRSV-positive samples collected in Xinjiang province in 2022. Through comprehensive genomic sequencing, phylogenetic analysis, and recombination analysis, we confirmed that this strain belongs to the NADC30-like recombinant PRRSV. During pathogenicity tests in piglets, this strain exhibited moderate virulence, causing symptoms such as reduced appetite, persistent fever, and weight loss; however, no mortality cases were observed. Tests conducted at various time points detected the presence of PRRSV nucleic acid in nasal swabs, rectal swabs, tissue samples, and blood, with the highest viral loads found in lung tissue and blood. Serum biochemical tests indicated significant impairment of liver and kidney function. PRRSV antibodies began to appear gradually after 10 days post infection. Hematoxylin and eosin staining revealed substantial pathological changes in lung tissue and lymph nodes. This study enhances our understanding of the epidemiology of PRRSV and underscores the importance of ongoing monitoring and research in light of the challenges posed by the continuous evolution of viral strains. Furthermore, the research emphasizes the urgency of the rapid genomic analysis of emerging viral strains. Through these comprehensive research and monitoring strategies, we aimed to curb the spread of PRRSV more effectively and thus reduce the huge economic losses it caused to the pig industry.

## 1. Introduction

Porcine reproductive and respiratory syndrome (PRRS) is one of the most economically significant diseases affecting pigs worldwide. First reported in the United States in 1987, PRRS rapidly spread across Europe and Asia within a few years. Based on geographic origins and genetic variations, PRRSV is classified into two distinct viral species: Betaarterivirus suid 1 (PRRSV-1, represented by the Lelystad virus) and Betaarterivirus suid 2 (PRRSV-2, represented by VR2322) [1,2]. In China, PRRS was first described in 1995, with the isolation of the PRRSV strain CH-1a. In 2006, a new type of PRRSV strain emerged, represented by strains such as JXA1 and TJ. Beginning in 2013, recombinant strains of PRRSV began to spread widely, with NADC30-like strains being the most prevalent [3,4]. PRRSV can be further categorized into eleven lineages based on the ORF5 gene. The epidemiological landscape of PRRSV in China is complex, currently classified into the lineages 1, 3, 5, and 8 of the PRRSV-2 type, with occasional occurrences of PRRSV-1. To date, lineage 1 remains the predominant strain domestically [5,6,7].

PRRSV is an enveloped, single-stranded positive-sense RNA virus with a genome approximately 15 kb in size, containing at least 10 open reading frames [8,9]. Notably, the PRRSV, ORF5, and NSP2 genes exhibit significant genetic diversity and are commonly utilized for typing and differential analysis [10,11]. PRRSV possesses a natural capacity for adaptation and evolution, resulting in considerable genetic and antigenic variability due to the error-prone nature of its RNA polymerase [9]. The virus primarily replicates in macrophages and dendritic cells in the lungs and upper respiratory tract, leading to viremia that can persist for several weeks. The lymphoid organs are the final replication sites before the virus is eliminated [12]. The epidemiological features of PRRSV are complex and variable, with transmission routes being highly widespread. Infected pigs can exhibit a range of clinical symptoms, from mild respiratory issues to severe reproductive problems and high mortality rates [13,14]. Given the high variability of PRRSV, the effectiveness of vaccines is often limited. Therefore, thorough monitoring of PRRSV and the development of effective prevention and control strategies are essential.

In China, the outbreak caused by the NADC30-like strain continues to occur, which initially only appeared in local areas, but has gradually spread to several provinces over time [15,16]. Due to the high genetic diversity and antigenic variability of the strain, the existing vaccine and prevention and control measures are much less effective in dealing with the virus, putting the pigs at a higher risk of infection. In the face of such a severe epidemic situation, we urgently need to deeply understand the biology, genetic characteristics, and pathogenic mechanisms of NADC30-like PRRSV strains circulating in different regions, which is of vital significance for the development of effective prevention and control strategies and the development of new vaccines.

This article details the successful isolation of a PRRSV NADC30-like strain in Xinjiang and analyzes its complete genome sequence. Additionally, pathogenicity tests were conducted to establish a solid foundation and scientific basis for epidemiological research and vaccine development related to PRRSV.

## 2. Materials and Methods

### 2.1. Ethical Statements

The animal experiments in this study were carried out according to the Chinese Regulations on Laboratory Animals. The protocol for primary PAM preparation and animal experiments was approved by the Biology Ethics Committee of Shihezi University, with approval no. A2024-434.

### 2.2. Clinical Sample Collection and Detection

In January 2022, a suspected case of PRRSV occurred on a small private pig farm in Xinjiang, characterized by clinical signs including stillbirths in sows, and respiratory symptoms in some fattening pigs. Meanwhile, some piglets also showed symptoms such as dyspnea and growth retardation. After obtaining permission from the animal owners, we collected serum and tissue samples from the affected pigs on the farm. Total RNA was extracted from the supernatant using the RNA Kit (TransGen, Beijing, China) and analyzed with PRRSV detection primers (Table 1). Primers for specific detection were designed for the NSP2 hypervariable region based on the GenBank accession PRRSV reference strain CH-1R (EU807840.1), TJ (EU860248.1) and NADC30 (JN654459.1).

### 2.3. Virus Isolation

Porcine alveolar macrophages (PAM) were obtained from 4-week-old PRRSV-negative pigs using a lung lavage technique and cultured in RPMI 1640 supplemented with 10% fetal bovine serum (FBS) and penicillin–streptomycin. All cells were cultured at 37 °C with 5% CO_2_. All cells were cultured at 37 °C with 5% CO_2_. PRRSV-positive serum samples were filtered through a 0.22 µm filter and inoculated into PAM in varying proportions for pathogen isolation. When the cells exhibited a cytopathic effect (CPE), the viral solution was collected by repeated freezing and thawing three times. At the same time, the Marc-145, MA-104, and NPTr cell lines were inoculated to observe the cytopathic effect.

### 2.4. Genome Sequencing

Primers for whole-genome sequencing were designed based on literature research [4] (Table 1). Total RNA was extracted from the supernatant using the RNA Kit (TransGen, Beijing, China), and reverse-transcribed into complementary DNA (cDNA) using the HiFiScript cDNA Synthesis Kit (CWBIO, Beijing, China). The PCR conditions included an initial denaturation at 95 °C for 5 min, followed by 35 cycles of denaturation (95 °C for 30 s), annealing (57.5 °C for 30 s), and extension (72 °C for 2 min), with a final extension at 72 °C for 10 min. Briefly, RT-PCR products were analyzed by 1% agarose gel electrophoresis. Target fragments were excised from the gels and purified using a TIANgel Midi Purification Kit (Tiangen, Beijing, China). The purified PCR products were then cloned into the vector pMD19-T (Takara, Dalian, China). Recombinant clones were sent to Ruibo Xingke Biotechnology Co., Ltd. (Beijing, China) for sequencing. Genome assembly and sequence alignments were performed using SeqMan and MegAlign in Lasergene (Version 7.1, DNASTAR Inc., Madison, WI, USA), respectively.

### 2.5. Phylogenetic Analysis

The ClustalW method using the DNAstar software was employed to analyze multiple sequence alignments. Phylogenetic analysis was conducted using the distance-based neighbor-joining method in MEGA (version 7.0), with bootstrap analysis performed using 1000 replicates. Phylogenetic trees for the PRRSV ZJ-Z5 and reference isolates (as shown in Appendix A) were constructed based on whole-genome and ORF5 gene sequences.

### 2.6. Recombination Analysis

Recombination analysis is essential for full-length genome analysis. All the genome alignment sequences were screened using the Recombination Detection Program 5 (RDP 5.0) using seven methods: including RDP, GENECONV, BootScan, MaxChi, Chimaera, SiCcan, and 3Seq [17]. Recombinant events were considered to be occurring if at least five of the seven methods had a *p*-value cut-off of 0.05. Subsequently, the putative segment exchanges were further verified using Simplot 3.5.1 software to analyze the genomic sequences, with a sliding window of 1000 bp (30 bp step size) [18].

### 2.7. Animals and Experimental Design

To evaluate the pathogenicity of the XJ-Z5 isolate, four 4-week-old piglets (PRRSV, PCV2, and PRV negative) were randomly divided into two groups of two piglets each. Piglets in one group were intramuscularly and intranasally inoculated with 2 mL of 10^5^ TCID_50_ per piglet, while in the other group they were inoculated with 2 mL of RPMI-1640 medium in the same manner. Body temperature and clinical signs were recorded daily, and body weight was recorded weekly. Whole blood, serum, nasal swabs, and fecal swabs were collected at 0, 4, 7, 10, 14, and 21 days post infection (dpi). The piglets survived until 21 dpi, at which point all piglets were euthanized for further pathological observations.

### 2.8. Viral Load Detection

After viral inoculation, oral, nasal, and rectal swabs, along with serum samples, are collected on different days. On the final day, tissues (heart, liver, spleen, lung, kidney, and lymph nodes) are also collected for testing. Total RNA is extracted using an RNA extraction kit (TransGen, Beijing, China). The extracted RNA is then reverse transcribed to synthesize cDNA. The viral load in oral and nasal swabs, rectal swabs, serum, and various tissues is quantified using real-time reverse transcription polymerase chain reaction (qRT-PCR). We established the absolute quantification of the viral load by standard plasmids. Primers used in qRT-PCR are shown in Table 1. The qRT-PCR reaction system consisted of 5 µL of SYBR Mixture (CWBIO, Beijing, China), 0.1 µL each of upstream and downstream primers, 3.8 µL of water, and 1 µL of template. The qRT-PCR program was as follows: 95 °C for 10 min, followed by 40 cycles of 95 °C for 10 s, 60 °C for 30 s, and 72 °C for 32 s.

### 2.9. Blood Biochemical

On day 21 post infection, a Celercare V5 Autobiochemical analyzer (MNCHIP, Tianjin, China) was used to determine Aspartate Aminotransferase (AST), γ-glutamyl transpeptidase (GGT) and creatinine (CRE) levels in serum.

### 2.10. Antibody Detection

On days 0, 4, 7, 10, 14, and 21 post infection, antibody titers were detected using the PRRSV N protein ELISA antibody detection kit (Keqian, Wuhan, China). A sample was considered positive for PRRSV N protein antibodies when the Kq value was greater than or equal to 0.4.

### 2.11. Histopathology

Lung and lymph node samples were collected during the autopsy. The tissue of the piglets was fixed in 4% paraformaldehyde for 48 h, and slices approximately 3 μm thick were prepared. The slices were then immersed twice in xylene for 10 min each. Absolute ethanol was used for dehydration. Lung and lymph node tissue samples were routinely stained with hematoxylin and eosin (Beyotime, Shanghai, China) for histopathological analysis.

### 2.12. Data Analysis

Using GraphPad Prism10.0 (San Diego, CA, USA) for data graphing and processing.

## 3. Results

### 3.1. Virus Isolation and Whole Genome Sequencing

Using specific PRRSV NSP2 primers based on PRRSV, HP-PRRSV, and the NADC30-like strain, the band showed 1021 bp when the sample was typical PRRSV, 931 bp for highly pathogenic PRRSV, and 628 bp corresponded to the NADC30-like strain. In this study, the sample showed a single band of about 628 bp (Figure 1A), which was initially determined as the NADC30-like strain. Sequencing showed that the sample was highly homologous to the NADC30-like strains. PCR was conducted according to the designed primers [4], successfully amplifying the entire genome of the XJ-Z5 strain in eight overlapping fragments. The negative control did not show amplification. The sizes of the target genes were 2167 bp, 1940 bp, 2146 bp, 2160 bp, 2199 bp, 2109 bp, 1744 bp, and 1623 bp, respectively. These sizes were consistent with the expected lengths, indicating more accurate results. Through Sanger sequencing of splicing and assembly fragments, we obtained a full-length genome measuring 15,015 nt (Figure 1B). The complete gene sequence was subsequently submitted to the GenBank database (GenBank number: PQ835040). Although no CPE was observed in Marc-145, MA-104, and NPTr cells, the XJ-Z5 virus was isolated in PAM cell (Figure 1C).

### 3.2. Results of Phylogenetic Analysis

To explore the genetic relationship between the XJ-Z5 strain and other PRRSV strains, we assembled the genome sequencing results of the isolated strains using DNASTAR software. We first performed a systematic evolutionary analysis of the ORF5 gene of XJ-Z5, which revealed that the 25 strains were distinctly divided into seven branches. Among these, the XJ-Z5 strain was found to be more distantly related to the PRRSV-1 type LV strain. Lineages 8.7 and 5.1 clustered together in one branch, while lineages 3 and 1.5 each formed separate branches, while XJ-Z5 clustered with lineage 1.8. Furthermore, a phylogenetic tree based on the whole-genome sequences of 24 representative Chinese reference PRRSV strains from different regions and years indicated that XJ-Z5 is clustered with NADC30-like isolates (Figure 2). The XJ-Z5 virus shares a common ancestor with the NADC30-like isolates, suggesting a close genetic relationship. Consequently, we concluded that the XJ-Z5 isolates are classified as NADC30-like PRRSV.

### 3.3. Results of Recombination Analysis

To identify whether recombination events occurred in the PRRSV XJ-Z5 strain, we analyzed the constructed genome-wide phylogenetic tree using RDP 5 and Simplot. Seven methods incorporated in RDP5 indicated that XJ-Z5 is a natural recombinant virus derived from five isolates, with HENAN-HEB and SD53-1603 serving as the major parental virus while JXA1 and HuN4 as the minor parental viruses (Table 2). The JXA1 isolate provides the 5708–6157 bp region encoding NSP4, and the HuN4 isolate provides the 10,881–11,241 bp region within NSP10-11, which aligns with the sequence alignment results (Table 2). Furthermore, multiple crossover events were confirmed using Simplot 3.5.1 (Figure 3). The JXA1 isolate provides the region from 5708 to 6157 bp, encoding NSP4, while HuN4 provides the region from 10,881 to 11,241 bp, which includes parts of the NSP10 and NSP11. These regions are consistent with the sequence alignment results (Table 3). In particular, the natural recombinant virus XJ-Z5 identified in this study is a recombination of two lineages of PRRSV2 (with HENAN-HEB and SD53-1603 belonging to lineage 1.8 and JXA1 and HuN4 belonging to lineage 8.7 [6]).

### 3.4. Results of Pathogenicity Analysis

The animal pathogenicity experiment procedure is shown in Figure 4A. Within a week of inoculation with PRRSV XJ-Z5, the piglets exhibited clear clinical symptoms, including loss of appetite, coughing, respiratory distress, and fever. In contrast, the control group did not display any significant clinical symptoms throughout the entire trial period. The body temperature of the challenge group began to rise from 6 dpi, exceeding 40 °C, and remained elevated until 19 dpi, while the body temperature of the control group remained within the normal range (Figure 4B). The average daily weight gain of the challenge group was significantly lower than that of the control group during the second and third weeks, with the challenge group experiencing negative growth, whereas the control group demonstrated steady growth (Figure 4C). Biochemical assessments of liver and kidney function conducted at 14 dpi revealed that the levels of AST, GGE, and GRE in the challenge group were significantly higher than the normal range (Figure 4D–F). That indicates that the liver and kidney function have received some degree of damage after PRRSV infection.

### 3.5. Viral Load and Antibody Levels

After conducting a necropsy on the heart, liver, spleen, lungs, and kidneys, the viral load analysis revealed that the viral content in the lung tissue was the highest, followed by the lymph nodes and spleen, and the lungs were significantly different from the heart, liver, and kidney in the challenge group (Figure 4G). Further testing of nasal and anal swabs indicated that during the first week of infection, there was a gradual decrease in viral load in the nasal swabs (Figure 4H), while the anal swab results showed a persistent viral presence (Figure 4I). In the infected group, a peak viremia was observed on day 10 (Figure 4J). Additionally, the levels of PRRSV antibodies in the infected group gradually increased from day 7, peaking on day 21 (Figure 4K), while a certain lag in antibody production was observed.

### 3.6. Results of Histopathology

Compared to the control group, significant pathological changes, such as consolidation and ecchymosis, were observed in the lung tissue of the challenged group (Figure 5A). The results of the HE staining revealed serous and hemorrhagic pneumonia, as well as interstitial lung enlargement, indicating interstitial pneumonitis (Figure 5B). In the lymph nodes, the experimental group exhibited pronounced lymphocyte necrosis. Additionally, we noted an increased presence of necrotic cell fragments and frequent karyorrhexis (Figure 5C).

## 4. Discussion

Since its initial discovery and domestic isolation in China in 1995, PRRSV has exhibited a long-term epidemic trend [6]. As an RNA virus and lacking the proofreading function of a 3′→5′ exonuclease [19], PRRSV has a high frequency of nucleotide substitutions and a rapid rate of variation. The domestic landscape of PRRSV is complex, with different regions exhibiting variations in the prevalent strains. Its extensive genetic variation and diversity are recognized challenges in the prevention and control of PRRS [5]. Genetic recombination in PRRSV, particularly between different subgenotypes and lineages, is a key mechanism for the emergence of new strains. Since 2013, PRRSV-2 strains similar to NADC30 have emerged in China, prompting extensive discussions regarding recombination events involving PRRSV-2 strains. The L1 strain is believed to have originated in the United States and is composed of NADC30-like and NADC34-like strains [20]. These events have frequently resulted in genetic exchange with previously widespread PRRSV strains in China.

Surveys conducted on the epidemiology of PRRSV in the central and eastern regions of China from 2016 to 2017 indicated that the predominant epidemic strains were recombinant viruses of the NADC30-like subtype and the JXA1 subtype [21]. Subsequently, between 2017 and 2018, the NADC30 strain emerged as one of the primary epidemic strains in Hubei Province [22]. By 2023, two new NADC30-like strains, isolated from Guangdong and Guangxi, were formed through the recombination of the NADC30-like strain with the HP-PRRSV JXA1 strain [23]. This discovery aligns with our research findings, which utilized NSP2 gene identification primers. Sequencing analysis revealed that the NSP2 region lacks a continuous amino acid sequence, indicating that the NADC30-like strain also persists in the Xinjiang region.

PRRSV exhibits significant cell tropism, and its mechanism of cell infection typically involves binding to cellular receptors [24]. During the infection process, primary alveolar macrophages in pigs become the primary target cells. African green monkey kidney epithelial cells, such as MA-104 and its derivatives, Marc-145, and CL2621, provide substantial support for PRRSV infection in vitro [25]. However, it is noteworthy that most NADC30-like strains can only survive in primary porcine alveolar macrophages, which may be related to the role of the scavenger receptor CD163 in facilitating the release and internalization of the virus within macrophages [25,26]. In this study, we successfully isolated a strain of PRRSV from the serum of diseased pigs. Initially, the authors attempted to directly infect Marc-145, NPTr, and MA-104 cells with the sample for virus isolation. However, after multiple attempts, it was determined that the sample exhibited low infectivity across various cell lines. Subsequently, the sample was adapted in PAM cells for 1 to 5 generations before attempting to infect other cell lines, but the results still indicated low infectivity (Figure 1C). Our results showed that the NADC30-like strain XJ-Z5 demonstrates poor infectivity in Marc-145 cells.

Genetic evolution analysis is an indispensable tool in studying viral evolution, and it has been widely applied in the genetic evolution research of PRRSV [27]. Phylogenetic analysis is helpful to understand the genetic relationship and evolutionary history of viruses. In general, the typing of PRRSV mainly depends on the ORF5 gene [28]. However, the variation in the ORF5 gene may not fully reveal the genetic diversity of PRRSV. Therefore, this study conducted a whole-genome sequence analysis of the XJ-Z5 strain. By comparing the whole genome sequences of strains from different sources, we found that XJ-Z5 is more closely related to the NADC30-like strains. Furthermore, the study also revealed the position of XJ-Z5 on the genetic evolutionary tree and its genetic distance from known strains. Through these analyses, we identified specific genetic markers for XJ-Z5, which helps to deeply understand the origin and transmission patterns of this strain. The results suggest that the genetic variations in XJ-Z5 may be related to specific geographic regions or host adaptability, providing a scientific basis for the development of targeted prevention and control measures. Future research should continue to focus on the trend of genetic variations in XJ-Z5 and how these variations affect the pathogenicity and transmission ability of the virus.

Genomic RNA recombination widely promotes the development of genetic diversity within viral populations and helps them adapt to changing environments and selective pressures. RNA viruses exhibit a high frequency of recombination phenomena, and the impact of recombinant strains on the host is dual [29]. RNA recombination in arteriviruses mainly occurs through homologous mechanisms. Typically, recombination occurs in genomic regions with high sequence homology, although there are differences in the location and frequency of recombination hotspots among different arteriviruses [30]. Sometimes, recombination hotspots are found in regions encoding non-structural proteins that are crucial for viral replication and transcription. However, regions encoding structural proteins can also be hot spots, which may change the structure of the virus and shape its immunogenic characteristics [31]. During the analysis of these recombination events, there is often an accompanying enhancement of viral adaptability and expansion of the host range. For example, recombination in PRRSV-2 strains may confer stronger transmission capabilities or better adaptability to specific hosts. Recombination hotspots are identified as the regions within the genome that have the highest rates and frequencies of recombination [31,32]. The determination of recombination positions is entirely based on the approximate positions obtained from the analysis of representative reference strains. Since the recombination process is random, the breakpoints are randomly distributed. However, the survival and replication of recombinant viruses may be influenced by natural selection pressures. Only a portion of recombinant viruses can survive, and the entry or replication ability of the recombinant virus population is enhanced, expanding during the process of viral proliferation [32]. Furthermore, recombination events may also lead to changes in viral virulence, which are reflected in clinical manifestations and pathological changes. In our study, we observed significant differences in pathogenicity between recombinant strains and traditional strains. The XJ-Z5 strain we isolated was moderately pathogenic to piglets. The piglets developed significant PRRSV-related symptoms after the challenge, including loss of appetite, weight loss, and elevated body temperature. By biochemical function, we also found that the liver and kidney received significant damage after XJ-Z5 infection. NADC30-like strains have a strong recombination potential. In the 28-day-old piglets, the NADC30-like Sichuan isolate also showed typical clinical symptoms of PRRSV, including viremia and long-term nasal detoxification [33]. After an in-depth analysis of the dynamic changes in viral load, we found that the viral content in lung tissue was the highest, and the lungs were significantly different from the heart, liver, and kidney in the challenge group. Viremia in the infected group peaked at day 10. This finding suggests that there may be temporal differences in the response of different tissues to the virus, which may be related to tissue-specific immune responses. The necropsy results also showed significant gross lesions in the challenge group (Figure 5). The HE staining revealed interstitial lung enlargement, and the lymph nodes exhibited pronounced lymphocyte necrosis (Figure 5). Combined with the genomic analysis, this revealed that the isolated strain shared significant similarities with the North American PRRSV-2 prototype strain, the NADC30-like strain, which has been associated with increased virulence and broader geographical distribution.

Furthermore, by comparing and analyzing the viral shedding data from nasal swabs and anal swabs, we revealed possible different roles of the respiratory tract and the gastrointestinal tract in the process of virus clearance. The rapid decline in viral shedding from nasal swabs may indicate that the respiratory epithelial cells have a rapid clear mechanism for the virus [34], while the persistence of viral shedding from anal swabs may be related to the complexity of the intestinal environment, suggesting that the intestine may become a long-term reservoir for the virus [35]. This is consistent with the report that NADC30-like PRRSV causes intestinal infections and tropism in piglets [36] and also provides us with ideas for considering the gut axis in subsequent studies.

In terms of antibody response, PRRSV antibody levels appear more delayed and have lower antibody levels. The peak of antibody levels occurs in the third week post infection, which coincides with the peak of viremia, indicating that the body initiates an effective immune response after the peak of viremia. These results have potential guiding significance for vaccine development and the formulation of immunization strategies. Furthermore, we observed that although NADC30-like strains are not easily susceptible in vitro cell culture, their transmission speed and infection rate are relatively high in naturally infected pig populations. This suggests that NADC30-like strains may have unique transmission routes or host adaptability, which requires further research to clarify.

Pathogenicity tests were performed on the isolated strain to assess its impact on pigs. These tests involved the inoculation of the virus into a controlled group of pigs and closely monitoring clinical signs, viral shedding, and pathological changes. The results indicated that the NADC30-like strain had a high replication rate in vivo and caused severe respiratory symptoms and lesions, which are characteristic of PRRSV infection. Of course, this study also has certain limitations. The number of samples within each group for the piglet pathogenicity test is relatively small, failing to meet the ideal standard of at least three per group. This situation may weaken the statistical significance of the experimental data. By carefully sorting out and comparing the experimental data, we have found that even with a limited number of samples, the differences between the experimental group and the control group still exhibit a certain degree of disparity. This finding further strengthened our confidence in the experimental results. In future studies, increasing the sample size to verify these findings remains crucial. The findings from this study not only contribute to the understanding of the genetic diversity and pathogenicity of PRRSV but also highlight the need for continuous surveillance and the development of updated vaccines that can provide broader protection against emerging strains. The complete genome sequence of the XJ-Z5 strain has been deposited in public databases, making it available for further research and as a reference for diagnostic test development.

## 5. Conclusions

In summary, genomic sequence analysis of the newly isolated PRRSV strain from Xinjiang indicates a high degree of homology with the NADC30-like strain. Through pathogenicity tests, we found that the clinical symptoms caused by this strain in pig populations are similar to those of typical PRRSV infections, with moderate virulence. In conclusion, this study reveals that NADC30-like strains still exist in China’s Xinjiang province, and provides new insights into understanding their genetic characteristics and transmission mechanisms.

## Figures and Tables

**Figure 1 viruses-17-00379-f001:**
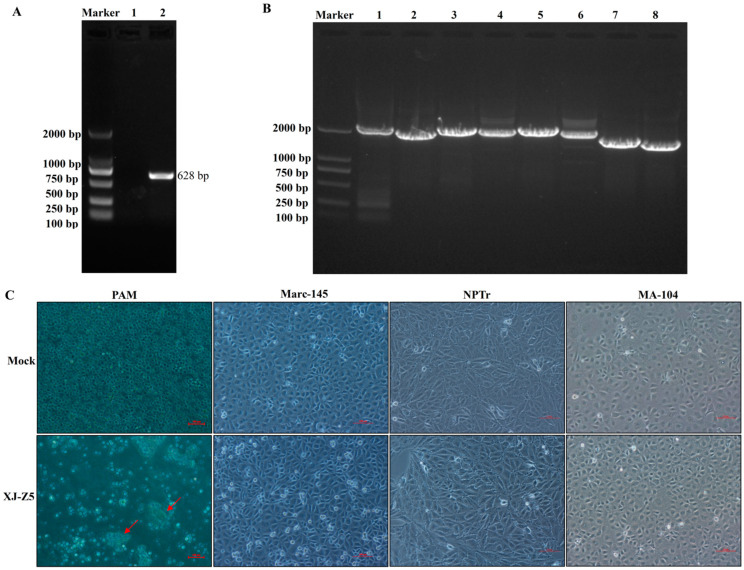
Detection and isolation of PRRSV XJ-Z5 strain. (**A**) RT-PCR identification of PRRSV NSP2 specific primers; (**B**) PRRSV XJ-Z5 strain whole-genome segment primer identification map; (**C**) PRRSV XJ-Z5 strain was inoculated on PAM, Marc-145, NPTr, and MA-104. The area indicated by the red arrow shows cytopathic effect.

**Figure 2 viruses-17-00379-f002:**
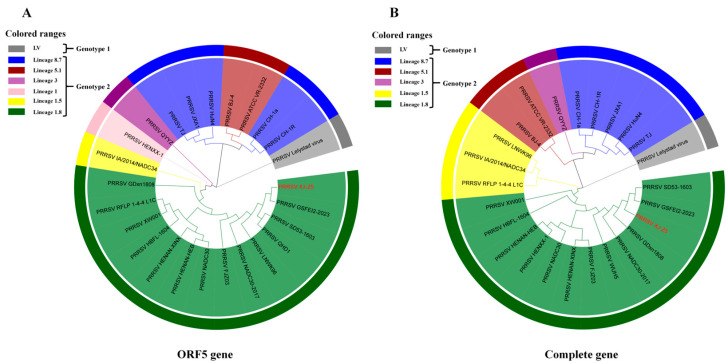
Phylogenetic analysis based on ORF5 and whole-genome. Phylogenetic trees were constructed based on ORF5 (**A**) and the complete genome (**B**) of the PRRSV XJ-Z5 strains with 24 representative Chinese reference PRRSV strains. Phylogenetic trees were constructed using the distance-based neighbor-joining method with 1000 bootstrap replicates in MEGA7. The red font highlights the strain PRRSV XJ-Z5 used in this study.

**Figure 3 viruses-17-00379-f003:**
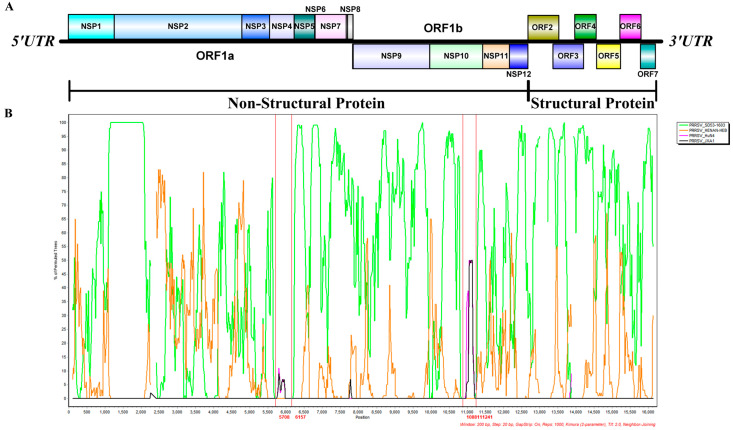
Recombination analysis. (**A**) Genomic diagram of the XJ-Z5 strain; (**B**) the crossover regions in the XJ-Z5 genome were further confirmed by Simplot 3.5.1. The crossover regions identified by Simplot were consistent with the results from the RDP5 analysis (Table 2). The *y*-axis shows the percentage of permutated trees employing a sliding window of 200 nucleotides (nt) and a step size of 30 nt. The other options, including the Kimura (2-parameter) distance model, 2.0 Ts/Tv ratio, neighbor-joining tree model, and 1000 bootstrap replicates were used.

**Figure 4 viruses-17-00379-f004:**
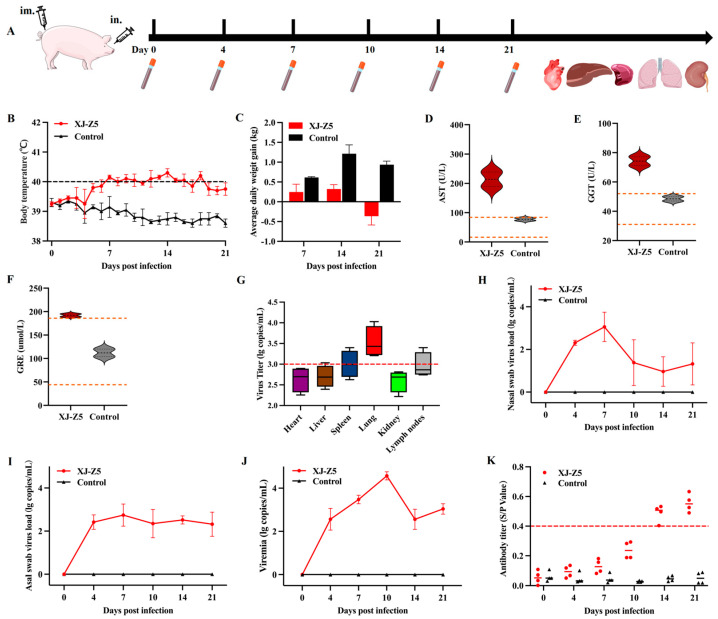
Pathogenicity results in piglets. (**A**) Animal challenge test procedure; (**B**) changes in body temperature after challenge; (**C**) changes in weight gain after challenge; (**D**–**F**) liver and kidney AST, GGT, and CRE biochemical indicators after challenge; (**G**) the viral load of heart, liver, spleen, lung, and kidney tissues after challenge; (**H**) the viral load of nasal swabs after challenge; (**I**) anal swab viral load after challenge; (**J**) viremia in the blood; (**K**) a PRRSV-specific antibody level was detected in each group during the challenge study.

**Figure 5 viruses-17-00379-f005:**
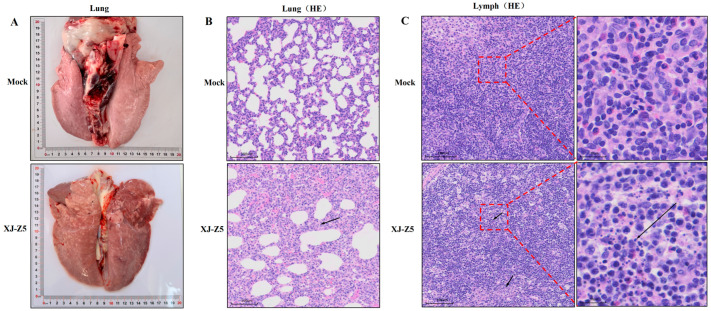
Lung autopsy and pathological sections after challenge infection. (**A**) Visual changes in lung anatomy; (**B**) results of lung hematoxylin–eosin staining. Scale bar = 100 µm. (**C**) Results of lymph node hematoxylin–eosin staining. The arrow indicates the characteristic area of the pathological changes. Scale bar = 100 µm and 20 µm.

**Table 1 viruses-17-00379-t001:** Primer sequence used in this study.

Primers	Sequences (5′-3′)	Amplification Position	Size (bp)
NSP2	F: ATGTTGTGCTTCCTGGGGTTG	2205–3225	628 (NADC30-like strain), 931 (HP-PRRSV), 1021 (PRRSV)
	R: CTTGACAGGGAGCTGCTTGA		
qRT-PCR	F: AAACCAGTCCAGAGGCAAGG	ORF 7	81
	R: TCAGTCGCAAGAGGGAATG		
P1	F: ATGACGTATAGGTGTTGGCTCTATGC	1–2167	2167
	R: AGCTTTCTCAAGCCTAGCCAAGC		
P2	F: GGGTTTGACCCTGCCTGCCTTGA	1823–3762	1940
	R: AAACTCACAAGCAGTGCCGACTG		
P3	F: CTGCTGGCTGGCTTTTGCTGTTG	3694–5839	2146
	R: CCTCCTTCCAGTTCGGGTTTGGC		
P4	F: CGCCCTTCAGGCCAGTTTTGTAA	5656–7816	2160
	R: CGCTAGGGGTCTTGTAAGGTATGTC		
P5	F: AGCGGCAAACCTGAACGGGTAAAAG	7741–9939	2199
	R: AAGCCTCAGGACATCAAGATGATTG		
P6	F: AAAGCTTTGGGCACGTGTCGGTTCA	9769–11,877	2109
	R: AACGGCAGGGCGCGGACGGAGTATC		
P7	F: CATCGCCGGATGGTTGGTGGTACTT	11,804–13,548	1744
	R: GCCATTCAGCTCACATATCGTCAGG		
P8	F: GATATGTTGGGGAAATGCTTGACCG	13,392–15,016	1623
	R: TTAATTACGGCCGCATGGTTCTC		

**Table 2 viruses-17-00379-t002:** Summary of crossover events in XJ-Z5 stains identified by RDP5.

Recombined Virus	Parental Virus		Breakpoints ^a^			Score for the Seven Detection Methods in RDP5 ^b^						
	Major	Minor	Region	Begin	End	RDP	GENECONV	BoostScan	MaxChi	Chimaera	SiScan	3Seq
XJ-Z5	HENAN-HEB	JXA1	NSP4	5708	6157	3.568 × 10^−33^	1.128 × 10^−35^	1.917 × 10^−23^	2.593 × 10^−10^	2.003 × 10^−9^	6.947 × 10^−11^	2.183 × 10^−3^
	SD53-1603	HuN4	NSP10~11	10,881	11,241	4.840 × 10^−23^	1.992 × 10^−18^	9.657 × 10^−24^	1.901 × 10^−11^	1.039 × 10^−8^	2.665 × 10^−2^	-

^a^ The breakpoints are based on the location in the genome of XJ-Z5. ^b^ The *p*-value cut-off is set at 0.05. *p* < 0.05 indicates the recombination events are significant.

**Table 3 viruses-17-00379-t003:** Nucleotide identity of XJ-Z5 compared with four PRRSV reference strains.

XJ-Z5	HENAN-HEB (%)	JXA1 (%)	SD53–1603 (%)	HuN4 (%)
Complete (15,015) ^a^	89.31	85.72	**92.07 ^b^**	85.5
ORF1a (191–7309)	87.21	84.10	**89.60**	84.18
NSP1(191–1336)	89.27	82.00	**88.84**	82.35
NSP2 (1337–4528)	85.45	82.34	**87.87**	82.25
NSP3 (4529–5221)	**90.32**	82.28	88.6	82.55
NSP4 (5222–5834)	81.32	**92.54**	85.15	92.18
NSP5 (5835–6346)	89.84	85.05	**92.59**	85.80
NSP6 (6347–6394)	85.88	**100.00**	95.83	99.46
NSP7 (6395–7171)	90.59	80.47	**94.59**	81.19
NSP8 (7172–7306)	88.89	88.15	**94.81**	86.67
ORF1b (7297–11,679)	92.4	87.00	**94.27**	86.86
NSP9 (7310–9221)	92.83	86.54	**95.71**	86.22
NSP10 (9226–10,548)	91.38	85.22	**93.88**	85.22
NSP11 (10,549–11,217)	**91.92**	90.87	91.02	91.02
NSP12 (11,218–11,679)	**94.63**	89.15	94.36	88.89
ORF2 (11,681–12,451)	91.51	86.57	**95.46**	86.57
ORF3 (12,304–13,068)	85.6	85.49	**95.18**	85.49
ORF4 (12,849–13,385)	93.3	84.57	**96.65**	84.92
ORF5 (13,396–13,995)	91.54	86.48	**95.67**	86.64
ORF6 (13,980–14,504)	93.33	88.59	**95.24**	88.59
ORF7 (14,494–14,865)	90.59	86.29	**94.09**	86.29

^a^ The location of each gene is based on the XJ-Z5 strain. ^b^ The segments of the reference strains with the highest similarity to XJ-Z5 are shown in bold.

## Data Availability

The datasets generated in this study are available upon request.

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
