# Peer review of "Genome and Pathogenicity Analysis of an NADC30-like PRRSV Strain in China’s Xinjiang Province"

_viruses, 2025, doi:10.3390/v17030379_

Round 1

Reviewer 1 Report

Comments and Suggestions for Authors

The manuscript by Li et al. describes the genomic and pathogenic features of a NADC30-like PRRSV isolated from Xinjiang in 2022. Even though many similar studies about NADC30-like PRRSV isolates from Chinese swine herds have been reported, the detailed characterizations of a Xinjiang isolate is still of great importance. However, there are still several issues should be addressed in this manuscript.

Major concerns:

1.     More background information about the XJ-Z5 isolate should be provided. The authors only mentioned that it’s isolated from serum collected from diseased pig. However, the infection information in that pig farm should be provided. How many pigs in that farm? What’s the morbidity and mortality? How many serum samples were collected and detected? The XJ-Z5 was isolated from diseased sow? Adult pig? Or piglet? Any other common pathogens detected in these serum samples?

2.     Figure 2 and figure 3 are too fuzzy. It’s impossible for the reviewer to evaluate which virus is closely related to XJ-Z5. Actually, the authors are recommended to perform complete genome and fragment comparisons between XJ-Z5 and other representative Chinese PRRSV isolates, which will facilitate the evaluation on the evolutionary relationships between XJ-Z5 and other Chinese isolates.

3.     Lines 196-199. The description is inconsistent with the results in Table 3. According to Table 3 and Figure 3, XJ/SHZ03 provides the 143-5745 bp fragment, JXA1 provides the 5746-6210 bp fragment, and HuN4 provides the 10919-11281 bp fragment, respectively. Please double check the results.

4.     The comparisons of genomic and pathogenic features between XJ-Z5 isolate and other reported Chinese NADC30-like isolates should be added.

5.     What’s the meaning of AST, GGE and GRE results? The authors should discuss and explain on the meaning of these results.

6.     Line 305, what are the specific genetic markers for XJ-Z5 identified in this study?

7.     Lines 335-336, lines 361-364, the authors did not provide any data in this manuscript to support these statements.

8.     Lines 384-385, “this study reveals the prevailing trend of NADC30-like strains in Xinjiang”, bases on the characterizations of only one NADC30-like PRRSV from Xinjiang (without an epidemiological investigation), how could this study reveal the prevailing trend of NADC30-like strains in Xinjiang?

9.     The animal experiment must be approved by an animal welfare and ethics committee. Please provide the ethics statement.

Author Response

Dear reviewer:

I am very grateful for your comments on the manuscript. According to your advice, we amended the relevant part of the manuscript. Here, we attached the revised manuscript with the corrected sections red-marked was attached in the formats of both PDF and MS Word, for your approval.

Some of your questions were answered below.

Comments 1: [More background information about the XJ-Z5 isolate should be provided. The authors only mentioned that it’s isolated from serum collected from diseased pig. However, the infection information in that pig farm should be provided. How many pigs in that farm? What’s the morbidity and mortality? How many serum samples were collected and detected? The XJ-Z5 was isolated from diseased sow? Adult pig? Or piglet? Any other common pathogens detected in these serum samples?]

Response 1: [In the "Clinical Sample Collection and Testing" section of the study methods section, We add to the relevant background information. The private farm has nearly 100 sows, is a small-scale farm, and has been vaccinated with other conventional vaccines. In this pig farm, typical PRRSV respiratory symptoms were mainly observed in lactating piglets and some fattening pigs during the suckling period, and a small number of piglets died. We aimed to detect and isolate the PRRSV strain, so ten piglets suspected of PRRSV infection and ten fattening pigs were randomly collected. Positive PRRSV results were found by testing PRRSV, PCV-2, and PRV only in the serum samples. Thus, the XJ-Z5 strain was isolated from the serum of diseased piglets. We also detected the tissues and organs of some dead piglets, which detected the PRRSV pathogen. In addition, some piglets occasionally had diarrhea symptoms. In this study, the pig farms were also tested for pig diarrhea virus, and the results were all negative. This part is not reflected in the manuscript.] 

Comments 2: [Figure 2 and figure 3 are too fuzzy. It’s impossible for the reviewer to evaluate which virus is closely related to XJ-Z5. Actually, the authors are recommended to perform complete genome and fragment comparisons between XJ-Z5 and other representative Chinese PRRSV isolates, which will facilitate the evaluation on the evolutionary relationships between XJ-Z5 and other Chinese isolates.]

Response 2: [We accept the proposal to compare XJ-Z5 with other representative strains in China and improved the resolution of Figures 2 and 3.]

Comments 3: [Lines 196-199. The description is inconsistent with the results in Table 3. According to Table 3 and Figure 3, XJ/SHZ03 provides the 143-5745 bp fragment, JXA1 provides the 5746-6210 bp fragment, and HuN4 provides the 10919-11281 bp fragment, respectively. Please double check the results.]

Response 3: [We have checked the results and revised them accordingly.] 

Comments 4: [The comparisons of genomic and pathogenic features between XJ-Z5 isolate and other reported Chinese NADC30-like isolates should be added.]

Response 4: [We have made revisions accordingly in the results and discussion.] 

Comments 5: [What’s the meaning of AST, GGE and GRE results? The authors should discuss and explain on the meaning of these results.]

Response 5: [We have made revisions accordingly in the results and discussion.]

Comments 6: [Line 305, what are the specific genetic markers for XJ-Z5 identified in this study?]

Response 6: [We mainly determined that this strain belongs to the NADC30-like strain using specific Nsp2 detection primers and confirmed the consecutive amino acid sequence of Nsp by sequencing alignment.] 

Comments 7: [ Lines 335-336, lines 361-364, the authors did not provide any data in this manuscript to support these statements.]

Response 7: [We accept the suggestions and supplement the corresponding references in the manuscript.]

Comments 8: [Lines 384-385, “this study reveals the prevailing trend of NADC30-like strains in Xinjiang”, bases on the characterizations of only one NADC30-like PRRSV from Xinjiang (without an epidemiological investigation), how could this study reveal the prevailing trend of NADC30-like strains in Xinjiang?]

Response 8: [We have made the revisions accordingly.] 

Comments 9: [ The animal experiment must be approved by an animal welfare and ethics committee. Please provide the ethics statement.]

Response 9: [We have added to ethics statement in the Materials and Methods, and provided the animal ethics certificate and the approval number.] 

Reviewer 2 Report

Comments and Suggestions for Authors
  1. More references in the Introduction section are needed.
  2. Some background on the outbreak should be added to the Introduction to explain why the authors decided to isolate/characterize the PRRSV strain.
  3. Consider moving Table 2 to the supplementary materials.
  4. Table 1 – add references.
  5. Clinical signs were observed among finishing piglets at the farm. However, for your experimental inoculation study, you used 4-week-old piglets. The authors may want to provide more information on the outbreak and explain why this age group was chosen.
  6. Consider rephrasing “without a specific source of infection.”
  7. Line 139 – provide a reference for the PCR (or protocol).
  8. Line 158 – consider elaborating on these differential primers.
  9. CPE in cell culture can be associated with many pathogens and even non-infectious factors. Additionally, have the authors attempted to infect PAMs with the virus successfully isolated on MARC-145 cells? This is especially important in the context of your discussion (pp. 283–286). The lack of PAM tropism (even in vitro) may be an important feature of this new virus. To assure readers that all procedures were performed correctly, consider infecting PAMs from the batch used for the experiment with another (or prototype) NADC30-like strain.
  10. I strongly recommend testing the obtained samples from experimental inoculations for their ability to induce CPE in PAMs.
  11. How many passages of the isolated virus were conducted? What were the titers?
  12. Did the authors use any monoclonal antibodies to stain infected MARC-145 cells?
  13. Figure 4: Double-check if there is a statistically significant difference between groups/organs. I believe there is a statistically significant difference between different time points (Fig. 4K).
  14. For more accurate results, consider assessing viral titers by titrating tissue/serum samples in MARC-145 cells.
  15. Lines 287–293: This information is very important and must be supported by appropriate data. Additionally, the data in the Results and Discussion sections seem confusing (it is unclear what the history of the isolated strain was before it was used for experimental inoculation).
  16. In the Discussion, try to highlight the major findings of the in vivo experiment.
  17. Did you find anything uncommon compared to the normal behavior of NADC30-like strains?
  18. Did the clinical signs observed in the experimental study match those from the outbreak?

Author Response

Dear reviewer:

Thank you very much for your comments on the manuscript. Based on your comment and request, we have made extensive modifications to the original manuscript. Here, we attached the revised manuscript with the corrected sections red-marked in the formats of both PDF and MS Word, for your approval.

Some of your questions were answered below. 

Comments 1: [More references in the Introduction section are needed.]

Response 1: [We have made the revisions accordingly.] 

Comments 2: [Some background on the outbreak should be added to the Introduction to explain why the authors decided to isolate/characterize the PRRSV strain.]

Response 2: [We have added this part to the introduction.]

Comments 3: [Consider moving Table 2 to the supplementary materials.]

Response 3: [We agree to move Table 2 to the supplementary material.] 

Comments 4: [More references in the Introduction section are needed.]

Response 4: [We have made the revisions accordingly.] 

Comments 5: [Clinical signs were observed among finishing piglets at the farm. However, for your experimental inoculation study, you used 4-week-old piglets. The authors may want to provide more information on the outbreak and explain why this age group was chosen.]

Response 5: [We have re-supplemented the relevant background information of the pig farm. The farm mainly showed PRRSV related symptoms during the weaning to nursery period. In view of this, we isolated positive samples from piglets with serum of nearly 28 days of age, so four-week-old piglets were selected as the study subjects.]

Comments 6: [Consider rephrasing “without a specific source of infection.”]

Response 6: [We have redescribed it. "without a specific source of infection" mainly refers to the experimental animals that were negative for PRRSV, PCV 2 and PRV pathogens.] 

Comments 7: [Line 139 – provide a reference for the PCR (or protocol).]

Response 7: [We have added and modified the PCR procedures in the original manuscript.]

Comments 8: [Line 158 – consider elaborating on these differential primers.]

Response 8: [We have made the revisions accordingly.] 

Comments 9: [CPE in cell culture can be associated with many pathogens and even non-infectious factors. Additionally, have the authors attempted to infect PAMs with the virus successfully isolated on MARC-145 cells? This is especially important in the context of your discussion (pp. 283–286). The lack of PAM tropism (even in vitro) may be an important feature of this new virus. To assure readers that all procedures were performed correctly, consider infecting PAMs from the batch used for the experiment with another (or prototype) NADC30-like strain.]

Response 9: [We attempted to isolate the original samples with MARC-145 cells and deinfected cell lines using viruses after passage on PAM with less satisfactory results. We have carefully revised some parts of the discussion.] 

Comments 10: [I strongly recommend testing the obtained samples from experimental inoculations for their ability to induce CPE in PAMs.]

Response 10: [After the animal experiment, we tried to isolate XJ-Z5 from tissues and serum and found that the CPE in PAM cells was significant, but not in CPE in other conventional cell lines.]

Comments 11: [How many passages of the isolated virus were conducted? What were the titers?]

Response 11: [The virus was passed down for a total of six generations and then purified and concentrated measuring TCID50 as 10^6.35/0.1mL.] 

Comments 12: [Did the authors use any monoclonal antibodies to stain infected MARC-145 cells?]

Response 12: [We did not do this part of the work. Mainly because XJ-Z5 shows low infectivity on MARC145 cells, the fluorescence quantification can only detect a particularly low copy number, which may remain the residual virus at the time of infection, and the PCR results are almost negative.] 

Comments 13: [Figure 4: Double-check if there is a statistically significant difference between groups/organs. I believe there is a statistically significant difference between different time points (Fig. 4K).]

Response 13: [We accepted the suggestion to do the data analysis and updated Figure 4D and Figure 4K.]

Comments 14: [For more accurate results, consider assessing viral titers by titrating tissue/serum samples in MARC-145 cells.]

Response 14: [We attempted to amplify and multiply the XJ-Z5 strain using the MARC-145,3D4 / 21 and NPTr cell lines. However, the results were not satisfactory and despite detecting the presence of the virus, the passage effect was not ideal. Similar to most NADC30 strains, strain XJ-Z5 was successfully passaged only on PAM cells, and the cytopathic effect (CPE) was evident. In infected animals, we assessed the virus titers by PAM cells. In addition, we also tried to isolate XJ-Z5 in tissues and serum after the animal experiment, and the results also showed that CPE was obvious in PAM.] 

Comments 15: [Lines 287–293: This information is very important and must be supported by appropriate data. Additionally, the data in the Results and Discussion sections seem confusing (it is unclear what the history of the isolated strain was before it was used for experimental inoculation).]

Response 15: [We have modified this in the Materials and Methods, Results and Discussion.]

Comments 16: [In the Discussion, try to highlight the major findings of the in vivo experiment.]

Response 16: [We have revised it in the discussion following the recommendations.] 

Comments 17: [Did you find anything uncommon compared to the normal behavior of NADC30-like strains?]

Response 17: [We observed that the clinical isolate caused diverse and complex disease symptoms and rapid rates of transmission. Both animals showed significant clinical symptoms almost simultaneously and had common dyspnea problems, convulsions, and standing instability in addition to the second week of infection. Diarrhea was occasionally seen at two weeks of infection, but we could not directly affirm related to PRRSV infection and we tested PEDV and PRoV results negative, which may also be associated with the decrease of immunity after PRRSV infection.] 

Comments 18: [Did the clinical signs observed in the experimental study match those from the outbreak?]

Response 18: [Through experiments on the pathogenic nature of 28-day-old piglets, we observed that they could cause typical symptoms of PRRS. The experimental animals showed significant loss of appetite and body weight loss, which almost coincided with the symptoms observed in piglets during clinical sampling in the field of pigs, and the pathological changes of interstitial pneumonia appeared in the lungs. However, due to our limited experimental conditions, the abortion data could not be compared with clinical sampling.] 

Round 2

Reviewer 1 Report

Comments and Suggestions for Authors

Major concerns:

  1. According to Table 2 and Figure 3, the description about the recombination events (Lines 224-227) are still wrong. The small cross-over fragments are from minor parental viruses rather than the major parental viruses.
  2. To support the recombination results, the reviewer would recommend the authors to add another table to show the complete genome and fargment sequence comparisons between XJ-Z5 and corresponding reference strains.
Comments on the Quality of English Language

Inconsistent spellings and grammar errors could be noticed throughout the manuscript. For instance, "Marc-145 and MARC145", "thereby mitigating the substantial economic losses it inflicts on the pig industry".

Author Response

Dear reviewer:

I am very grateful for your comments on the manuscript. Per your advice, we continue to make serious revisions based on the first revision. The second revision is marked in blue, and we will attach the PDF and MS Word format for your approval.

Comments 1: [According to Table 2 and Figure 3, the description about the recombination events (Lines 224-227) are still wrong. The small cross-over fragments are from minor parental viruses rather than the major parental viruses]

Response 1: [Following your recommendation, this section of the description of the results has been meticulously reviewed and verified using the software RDP 5 and Simplot. Subsequently, these results have been redescribed in the revised manuscript.] 

Comments 2: [To support the recombination results, the reviewer would recommend the authors to add another table to show the complete genome and fargment sequence comparisons between XJ-Z5 and corresponding reference strains]

Response 2: [We accept your recommendation and add Table 3 XJ-Z5 to compare with the four PRRSV reference strains. The results of Table 3 were also added to the Recombination Analysis.]

Comments on the Quality of English Language: [Inconsistent spellings and grammar errors could be noticed throughout the manuscript. For instance, "Marc-145 and MARC145", "thereby mitigating the substantial economic losses it inflicts on the pig industry".]

Response : [Following your suggestion, we have carefully reviewed the spelling and grammar of the full text and made unified modifications.]

Reviewer 2 Report

Comments and Suggestions for Authors

While the majority of the comments have been addressed, it remains unclear whether the corresponding corrections have been incorporated into the manuscript. In your response, please specify where (line number) the corrections can be found.

Author Response

Dear reviewer:

Thank you very much for your comments on the manuscript. Based on your comment and request, we have made extensive modifications to the original manuscript. Here, we will make the first revision in red and the second revision in blue, and provide the revised manuscript in PDF and MS Word format for your review and approval.

Some of your questions were answered below. 

Comments 1: [More references in the Introduction section are needed.]

Response 1: [We have made the revisions accordingly, adding more references to the Introduction. Details can be seen in the introduction to the second page of the revised manuscript.]

Comments 2: [Some background on the outbreak should be added to the Introduction to explain why the authors decided to isolate/characterize the PRRSV strain.]

Response 2: [We accept your advice and add this part to the Introduction, showing lines 72 to 80 in the revised manuscript. ]

Comments 3: [Consider moving Table 2 to the supplementary materials.]

Response 3: [We accept your recommendation to move Table 2 to the supplementary material and change Table 2 to Supplementary Table 1.]

Comments 4: [Table 1 – add references.]

Response 4: [We accepted your advice and made the revisions accordingly. We detailed the design of the detection primers in lines 98-100, the reference source of the genome primers P1-P8 is identified in line 113.]

Comments 5: [Clinical signs were observed among finishing piglets at the farm. However, for your experimental inoculation study, you used 4-week-old piglets. The authors may want to provide more information on the outbreak and explain why this age group was chosen.]

Response 5: [We have re-supplemented the relevant background information of the pig farm. Details are provided in lines 92-96 of the revised manuscript. The farm mainly showed PRRSV-related symptoms during the weaning to nursery period. Given this, we isolated positive samples from piglets with a serum of nearly 28 days of age, so four-week-old piglets were selected as the study subjects. ]

Comments 6: [Consider rephrasing “without a specific source of infection.”]

Response 6: [We have redescribed it. Details are provided in line 144 of the revised manuscript. "without a specific source of infection" mainly refers to the experimental animals that were negative for PRRSV, PCV 2, and PRV pathogens.]

Comments 7: [Line 139 – provide a reference for the PCR (or protocol).]

Response 7: [We have added and revised the PCR procedures in the revision of the manuscript. Details are provided in lines 158-161 of the revised manuscript.]

Comments 8: [Line 158 – consider elaborating on these differential primers.]

Response 8: [We have made corresponding modifications, showing lines 184-189 in the revised manuscript.]

Comments 9: [CPE in cell culture can be associated with many pathogens and even non-infectious factors. Additionally, have the authors attempted to infect PAMs with the virus successfully isolated on MARC-145 cells? This is especially important in the context of your discussion (pp. 283–286). The lack of PAM tropism (even in vitro) may be an important feature of this new virus. To assure readers that all procedures were performed correctly, consider infecting PAMs from the batch used for the experiment with another (or prototype) NADC30-like strain.]

Response 9: [XJ-Z5 showed significant PRRSV strong CPE on PAM cells and was strongly positive for PRRSV detection. XJ-Z5 was selected and purified by single-cell lesions for two passages, but it was less infectious on other cell lines. The results are not shown in the manuscript. We attempted to isolate the original samples with MARC-145 cells and deinfected cell lines using viruses after passage on PAM with less satisfactory results. We have carefully revised some parts of the discussion, showing lines 328-335 of the revised manuscript.]

Comments 10: [I strongly recommend testing the obtained samples from experimental inoculations for their ability to induce CPE in PAMs.]

Response 10: [We did this part of the work, after the animal experiment, we tried to isolate XJ-Z5 from tissues and serum and found that the CPE in PAM cells was significant, but not in CPE in other conventional cell lines. This part of the work is not represented in the manuscript.]

Comments 11: [How many passages of the isolated virus were conducted? What were the titers?]

Response 11: [The virus was passed down for a total of six generations and then purified and concentrated measuring TCID50 as 10^6.35/0.1mL.]

Comments 12: [Did the authors use any monoclonal antibodies to stain infected MARC-145 cells?]

Response 12: [We did not do this part of the work. Mainly because XJ-Z5 shows low infectivity on MARC145 cells, the fluorescence quantification can only detect a particularly low copy number, which may remain the residual virus at the time of infection, and the PCR results are almost negative.]

Comments 13: [Figure 4: Double-check if there is a statistically significant difference between groups/organs. I believe there is a statistically significant difference between different time points (Fig. 4K).]

Response 13: [We accepted the suggestion to do the data analysis and updated Figure 4D and Figure 4K. Our results are mainly by comparing data at the same time point, focusing on trends.]

Comments 14: [For more accurate results, consider assessing viral titers by titrating tissue/serum samples in MARC-145 cells.]

Response 14: [We attempted to amplify and multiply the XJ-Z5 strain using the MARC-145,3D4 / 21 and NPTr cell lines. However, the results were not satisfactory and despite detecting the presence of the virus, the passage effect was not ideal. Similar to most NADC30 strains, strain XJ-Z5 was successfully passaged only on PAM cells, and the CPE was evident. In infected animals, we assessed the virus titers by PAM cells. In addition, we also tried to isolate XJ-Z5 in tissues and serum after the animal experiment, and the results also showed that CPE was obvious in PAM.]

Comments 15: [Lines 287–293: This information is very important and must be supported by appropriate data. Additionally, the data in the Results and Discussion sections seem confusing (it is unclear what the history of the isolated strain was before it was used for experimental inoculation).]

Response 15: [We added additional pictures of NPTr and MA-104 cells in Figure 1C with modifications in the methods, results, and discussion, as detailed in lines 109-111, 196-198, and 328-335. Our initial purpose was to isolate PRRSV strains with Xinjiang characteristics, to provide an animal model basis for our other vaccine research and challenge experiments.]

Comments 16: [In the Discussion, try to highlight the major findings of the in vivo experiment.]

Response 16: [We accept your recommendation to revise the Discussion. The findings were elaborated and discussed, with added references, as detailed in lines 376-404.]

Comments 17: [Did you find anything uncommon compared to the normal behavior of NADC30-like strains?]

Response 17: [According to the literature on NADC30-like strain's pathogenicity, we observed our clinical isolate caused diverse and complex disease symptoms and rapid rates of transmission. Both animals showed significant clinical symptoms almost simultaneously and had common dyspnea problems, convulsions, and standing instability in addition to the second week of infection. Diarrhea was occasionally seen at two weeks of infection, but we could not directly affirm related to PRRSV infection and we tested PEDV and PRoV results negative, which may also be associated with the decrease of immunity after PRRSV infection. This also suggests that we should increase the number of experimental animals in other subsequent vaccine challenge experiments to more accurately assess the vaccine effect and the response of the animals.]

Comments 18: [Did the clinical signs observed in the experimental study match those from the outbreak?]

Response 18: [Through experiments on the pathogenic nature of 28-day-old piglets, we observed that they could cause typical symptoms of PRRS. The experimental animals showed significant loss of appetite and body weight loss, which almost coincided with the symptoms observed in piglets during clinical sampling in the field of pigs, and the pathological changes of interstitial pneumonia appeared in the lungs. However, due to our limited experimental conditions, the abortion data could not be compared with clinical sampling.]

Round 3

Reviewer 2 Report

Comments and Suggestions for Authors

All comments have been addressed

Author Response

Dear Editor:

 I am very grateful for your comments on the manuscript. Following your suggestions, we have made careful revisions and marked them in red. We will attach the PDF and MS Word formats for your approval.

Comments 1: [In the title, “in Xinjiang China” is suggested to corrected as “in Xinjiang province of China”]

Response 1: [We accept your suggestion and made modifications to the title.] 

Comments 2: [In the abstract, line 18 “Porcine Reproductive and Respiratory Syndrome Virus“ is suggested to corrected as “porcine reproductive and respiratory syndrome virus”. ]

Response 2: [We accept your suggestion and have made the corresponding modifications.]

Comments 3: [Line 24, “in Xinjiang” is suggested to corrected as “in Xinjiang province”]

Response 3 : [We accept your suggestion and have made the corresponding modifications.]

Comments 4: [Line 26, “NADC30-like recombinant lineage”, please correct it to “NADC30-like recombinant PRRSV”]

Response 4: [We accept your suggestion and have made the corresponding modifications.]

Comments 5: [Line 40 and line 44, “Porcine reproductive and respiratory syndrome”, only the first letter of the sentence is capitalized.]

Response 5: [We accept your suggestion and have made the corresponding modifications.]

Comments 6: [Line 49, please delete the word “virus” after “VR-2332”.]

Response 6: [We accept your suggestion and have made the corresponding modifications.]

Comments 7: [Line 49 and line 54, “In our country” is suggested to corrected as “In China”]

Response 7: [We accept your suggestion and have made the corresponding modifications.]

Comments 8: [Line 53, “nine lineages” is suggested to corrected as “eleven lineages”]

Response 8: [We accept your suggestion and have made the corresponding modifications.]

Comments 9: [Line 57, “Porcine reproductive and respiratory syndrome virus (PRRSV)”should be “PRRSV”, since the abbreviation has been given in the first paragragh.]

Response 9: [We accept your suggestion and have made the corresponding modifications.]

Comments10: [Line 92, “outbreak” is suggested to corrected to “case”. “stillbirths, and respiratory diseases in some fattening pigs” is suggested to revise as “stillbirths in sows, and respiratory symptoms in some fattening pigs”]

Response 10: [We accept your suggestion and have made the corresponding modifications.]

Comments 11: [Line 100, deletion the “-“ in the “NADC-30”. The correct word is “NADC30”]

Response 11: [We accept your suggestion and have made the corresponding modifications.]

Comments 12: [Line 103, the age of piglets should be definitive in a given experiment ]

Response 12: [We accept your suggestion and have made the corresponding modifications.]

Comments 13: [Line 108, please delete the word “noticeable”]

Response 13: [We accept your suggestion and have made the corresponding modifications.]

Comments 14: [Line 127, “the DNA-STAR software” is suggested to corrected to “the DNAstar software”  DNAstar ]

Response 14: [We accept your suggestion and have made the corresponding modifications.]

Comments 15: [Line 133, Supplementary Table 1, some information is not correct, such as the “Lelystad virus M96262.1 USA”, “HuN4 EF635006.1 Henan, China” are not correct.  The “QHD1 MG687491.1 Hebei, China” is suggested to be double-checked. The “19 NADC30-like” is not suitable to be shown since it is lack of an exact name.]

Response 15: [We accept your suggestion, we have verified the supplementary Table 1 and made the corresponding modifications.]

Comments 16: [Line 143-145, the number of piglets is too small. In general, the number of piglets in each group should be more than three, and the experimental data is not statistically significant until the number is more than three. Please discuss it in the Discussion section. ]

Response 16: [We sincerely appreciate you pointing out the issue. At the initial stage of the experimental design, due to various objective constraints such as limitations of the experimental site and difficulties in selecting negative piglets, the number of piglets in each group failed to meet the ideal standard of three or more. We are well aware of the importance of sample size for the statistical significance of experimental data, and therefore, in the discussion section of the paper, we have supplemented the analysis of the potential impact of insufficient sample size on the experimental results.]

Comments 17: [Line 143 “piglets without a  specific source of infection(PRRSV, PCV2, and PRV negative)” is suggested to revise as “piglets (PRRSV, PCV2, and PRV negative)”]

Response 17: [We accept your suggestion and have made the corresponding modifications.]

Comments 18: [Line 146, “10^5 TCID50 virus particles”, the “10^5” should be revised in a correct form. The “virus particles” is suggested to be replaced with “per piglet” ]

Response 18: [We accept your suggestion and have made the corresponding modifications.]

Comments 19: [Line 150, “days post-challenge” and “days post-infection” should be uniform.]

Response 19: [We accept your suggestion and have made the corresponding modifications.]

Comments 20: [Line 154, “various tissues……” is suggested to be revised as “tissues (lung, liver……)…..”]

Response 20: [We accept your suggestion and have made the corresponding modifications.]

Comments 21: [Line 158, the weight or volume of samples should be uniform in this study and given in this manuscript, since an absolute quantification method was employed in the study.]

Response 21: [We accept your suggestion and have made the corresponding modifications.]

Comments 22: [Line 179-181, there are only two piglets in each group, the significant differences between the two groups is unconvincing.]

Response 22: [We accept your suggestion and have made the corresponding modifications. We have removed the statistical analysis.]

Comments 23: [Line 184, deletion the word “identification”. ]

Response 23: [We accept your suggestion and have made the corresponding modifications.]

Comments 24: [Line 198, deletion the word “successfully”. ]

Response 24: [We accept your suggestion and have made the corresponding modifications.]

Comments 25: [Figure 1, the letter size is too small to easily read. It is recommended to enlarge the size.]

Response 25: [We accept your suggestion and have made the corresponding modifications.]

Comments 26: [Line 202-204, the full-name of cell lines is recommended to be removed from the figure legend ]

Response 26: [We accept your suggestion and have made the corresponding modifications.]

Comments 27: [Line 208, “GP5” is suggested to be revised as “ORF5” ]

Response 27: [We accept your suggestion and have made the corresponding modifications.]

Comments 28: [Line 213, 221, “China” is suggested to be revised as “Chinese” ]

Response 28: [We accept your suggestion and have made the corresponding modifications.]

Comments 29: [Line 226-234, the recombination analysis result is suggested to be double-checked, and draw a conclusion carefully. The HuN4 and JXA1 is high identity, both strains can be considered as one strain in viral recombination analysis.]

Response 29: [We have accepted your suggestion and reviewed the results of the restructured software analysis. According to the score from RDP5 software, the restructured results are consistent with the manuscript. The two major parents are different, with the minor parent JXA1 providing the region from 5708 to 6157bp, encoding NSP4, while the minor parent HuN4 provides the region from 10881 to 11241bp, which includes parts of NSP10 and NSP11.]

Comments 30: [Fig3A, the genomic diagram of PRRSV is not correct. Fig 3A and Fig 3B should be the same width. ]

Response 30: [We accept your suggestion and have made the corresponding modifications. We have redrawn the structural diagram of Figure 3A and adjusted the widths to be consistent.]

Comments 31: [Fig4, the significant differences between the two groups is suggested to be removed from the figure.]

Response 31: [We accept your suggestion and have made the corresponding modifications. We have removed all statistical analyses from Figure 4.]

Comments 32: [Fig4C, the word “Contro” is suggested to be corrected.]

Response 32: [We accept your suggestion and have made the corresponding modifications. Modify the caption of Figure 4C correctly.]
